# OpenReview forum: "Hypothesis Hunting with Evolving Networks of Autonomous Scientific Agents"
_ICLR.cc/2026/Conference — Submitted to ICLR 2026_

### Official Review · Reviewer_rEBh · 2025-10-15

**Soundness:** 3
**Presentation:** 3
**Contribution:** 3
**Rating:** 6
**Confidence:** 3

**Summary:**

The paper presents "Hypothesis Hunting," an AI-driven paradigm for autonomous scientific discovery on large-scale datasets (e.g., genomics, Earth systems). It formalizes discovery in the AScience framework—modeling dynamic interactions among agents, networks, and evaluation protocols—and implements ASCollab, a decentralized LLM-based agent network with heterogeneous expertise, ReAct reasoning, shared registries, and meta-review tournaments. Evaluated on TCGA, the system rediscovers known biomarkers (e.g., BIRC5), extends pathways (e.g., ferroptosis), and proposes novel targets (e.g., SLC5A2, ABCC8), outperforming baselines on novelty and quality metrics; emergent agent behaviors and self-evolving knowledge flows are observed, and a clinical case independently validates BIRC5 and PRKD1 in renal cancer while suggesting dual-target strategies and candidate drugs. Limitations include domain specificity and the need for wet‑lab validation, but the work demonstrates AI as a collaborative catalyst for bridging data-to-knowledge gaps.

**Strengths:**

1. Proposes the Hypothesis Hunting paradigm for AI-driven autonomous scientific discovery, overcoming the limits of preset questions.

2. Introduces the AScience theoretical framework that formalizes discovery as dynamic interactions among agents, networks, and evaluation protocols, and implements it via ASCollab, which integrates heterogeneous agents, self-organizing networks, and peer‑review mechanisms.

3. Implements a distributed agent system that leverages heterogeneous LLM agent behaviors and supports diverse exploration strategies. Uses a two‑tier review (expert review + meta‑review) and a shared knowledge registry to ensure hypothesis novelty and quality. Validated on TCGA cancer data: rediscovers known biomarkers (e.g., BIRC5), extends pathways (e.g., ferroptosis), and proposes new targets (e.g., SLC5A2, ABCC8).

4. Simulates competition–collaboration dynamics of scientific communities (e.g., reputation systems) to avoid premature convergence and enhance discovery diversity. Networked agents outperform independent agents, yielding more high‑quality and novel hypotheses.

5. Integrates multi‑omics data (transcriptomics, proteomics, clinical) with external knowledge bases (e.g., PubMed, DepMap) to improve the biological plausibility of proposed hypotheses.

**Weaknesses:**

1. Experimental results are based solely on TCGA cancer genomics data and have not been validated in other disciplines (e.g., physics, chemistry).
2. The approach assumes subsequent wet‑lab validation; the current system only provides candidate lists, and its actual biomedical value remains to be determined.
3. The distributed agent network and continuous review mechanisms impose substantial compute requirements.
4. Agent decision logic is largely a black box, which affects reproducibility of results.
5. The evolutionary mechanisms of the dynamic network structure (e.g., convergence efficiency) require further theoretical analysis.

**Questions:**

See weakness.

---

> ### Author Response · Authors · 2025-11-25
> **Response to Reviewer rEBh (Part 1/2)**
>
> *Thank you for your thoughtful and comprehensive review. We appreciate your recognition of the Hypothesis Hunting paradigm, the AScience framework, and the strengths of ASCollab’s social dynamics and biological findings. Below, we address each of your key concerns.*
>
> ---
>
> ### [P1] Evaluation on Additional Domains
>
> We agree that demonstrating generality beyond cancer genomics is essential. During the rebuttal period, we deployed `ASCollab` on two additional, scientifically distinct datasets, without modifying the architecture or social mechanisms:
>
> 1. **NHANES (Public Health)**: harmonized participant-level dataset (with demographic, clinical, lifestyle, exposure, clinical variables and myocardial-infarction outcome) commonly used in epidemiology research.
> 2. **MYC-perturbation multi-omics dataset (Proprietary)**: A specialized dataset integrating eNET-seq RNAPII occupancy, bulk RNA-seq expression, and TRRUST TF–target interactions under MYC activation/inhibition. This represents a somewhat niche, high-dimensional regulatory dataset not commonly encountered in LLM training corpora.
>
> **Results:** We presented detailed analysis in `App A2`. In both domains, the system successfully adapted its tool usage and research behaviors without architectural changes, generated several domain-plausible and non-obvious findings:
> * **NHANES — Acidosis × Insulin-Resistance Interaction**: ASCollab identified a synergistic interaction between low serum bicarbonate and high TyG index in elevating MI risk, with serum albumin moderating this effect, surfacing a cross-metabolic interaction not typically examined in standard epidemiologic workflows.
> * **NHANES — Liver–Kidney Metabolic Flexibility Marker:** The system proposed the AST/Creatinine ratio as a protective marker of metabolic flexibility with a non-linear effect, particularly in metabolic-syndrome subgroups, highlighting a physiologically interpretable but underutilized biomarker.
> * **MYC-Perturbation Dataset — MYC-Aligned Anabolic Program:** ASCollab identified a coherent regulatory triad (EEF2, SLC7A5, FASN) that together constitute a MYC-driven anabolic program, integrating transcriptional, pausing, and pathway data into a unified mechanistic model.
>
> **Takeaway.** When considered along with existing results, we have demonstrated `ASCollab` operating over seven distinct modalities: RNA-Seq, proteomics, pathway/topology data, drugs & perturbations databases, Net-Seq, and population-level epidemiology. This breadth highlights that the system is **modality-agnostic and its social architecture transfers effectively to new scientific regimes**.
>
> **Note #1.** Due to limited expert availability during the rebuttal period, we provide qualitative domain-expert assessments rather than full rubric-based scoring. Nevertheless, in both datasets ASCollab produced novel, high-quality, and domain-plausible hypotheses. We will include expanded expert evaluations in the camera-ready version.
>
> **Note #2.** Our choice of additional evaluations is constrained by domain expertise, and we acknowledge that `ASCollab` has not yet been systematically tested in non-biological domains. While such studies are difficult to conduct within the rebuttal period, we now explicitly highlight this as an important direction for future work. We remain confident that the framework’s core principles are broadly domain-agnostic.
>
> ---
>
> ### [P2] Addressing Limitations
>
> **(1) Wet-lab validation.**
> We fully agree that assessing the scientific value of the discovered hypotheses requires wet-lab validation. As noted in the paper, this is a current limitation given the substantial cost and infrastructure required, which are beyond the resources of our academic lab. We have now made this clearer in the Introduction and Discussion. Our goal in this work is to show that ASCollab can generate plausible and mechanistically grounded hypotheses worthy of downstream investigation. In this regard, expert evaluations have confirmed this, and several high-scoring hypotheses were later partially validated by independently published studies.
>
> **(2) Compute requirements.**
> ASCollab’s distributed agents and iterative discovery cycles incur higher compute cost than simpler agentic baselines. However, both our new ablations (**[P3]**) and existing analyses show that these social mechanisms substantially improve novelty, quality, and diversity. We believe that the compute overhead is offset by the potentially higher scientific value produced by the system. We explicitly highlight this trade-off as a limitation in the revision.

---

> > ### Author Response · Authors · 2025-11-25
> > **Response to Reviewer rEBh (Part 2/2)**
> >
> > **(3) Reproducibility and black-box decision logic.**
> > While LLM decision-making is not fully transparent, the workflow is fully reproducible. Each agent outputs executable code, analysis logs, and review/meta-review documents, all stored in the registry. Re-running the generated code enables reproduction of quantitative results exactly, and the review chain provides a human-interpretable record of how evaluations and decisions were made.
> >
> > ---
> >
> > ### [P3] Additional Analysis of Network Dynamics
> >
> > This is an insightful point. More detailed analysis of how specific social mechanisms influence network behavior and frontier coverage would further strengthen the framework.
> >
> > To this end, we conducted new ablation studies that isolate the contribution of four key mechanisms: (1) endogenous collaboration, (2) heterogeneity in agents' epistemic behavior, (3) shared social memory, and (4) distributed agent populations. The ablation configurations are summarized below:
> >
> > | Ablation | Collaborations | Heterogeneous | # agents / distributed |
> > |---|---|---|---|
> > | single-agent (GPT-5) | N/A | N/A | 1 / ❌ |
> > | independent_homogeneous | ❌ | ❌ | 16 / ✔️ |
> > | collaborating_homogeneous | ✔️ | ❌ | 16 / ✔️ |
> > | ASCollab | ✔️ | ✔️ | 16 / ✔️ |
> >
> > **Key takeaways.** We refer you to `App A1`, which presents detailed analyses. Here, we describe the key findings in brief:
> > 1. **ASCollab produces the strongest hypotheses in direct head-to-head tournaments.** On KIRC, human experts compared the n-th best paper from each ablation in a four-way tournament (thus giving n=25 tournaments). Across all tournaments, ASCollab’s submissions were judged most scientifically significant, outperforming both `collab_homo` and the single powerful GPT-5 agent (`Fig 6a`).
> > 2. **Collaboration (even without heterogeneity) substantially improves quality-novelty.** Compared to `ind_homo`, the `collab_homo` condition produced higher-quality and higher-novelty findings, reflecting the value of endogenous collaboration in consolidating high-value findings and surfacing promising regions for further exploration.
> > 3. **Epistemic heterogeneity is a driver of novelty and diversity.** Comparing `collab_homo` vs. `ASCollab` (collaborative but heterogeneous agents), the heterogeneous population produced hypotheses that were more novel, conceptually broader, and explored more distinct research directions (`Fig 6c`).
> > 4. **A strong single agent is competitive—but fundamentally limited.** While the GPT-5 single agent generated high-quality hypotheses, its findings were much less diverse, clustering around a narrower conceptual space (as shown in `Fig 6b`). Without distributed and collaborative exploration, it cannot cover the novelty–quality–diversity frontier reached by the collective system.
> >
> > In addition to the findings above (which were regarding scientific outcomes produced by the system), we also analyzed how heterogeneity and collaboration affects individual agent behavior and network evolution.
> > * Our **agent-behavior analysis** (`Fig 7`) shows that heterogeneous agents develop **distinct methodological niches**, differ widely in code generation and tool use, and pursue different exploration–exploitation strategies. (In contrast, homogeneous agents behave nearly identically.)
> > * **Network-evolution analysis** (`Fig 8`) reveals that ASCollab exhibits higher connectivity, more cross-cluster interaction, lower influence inequality, and greater cumulative exchange of ideas compared to homogeneous collaborators. This demonstrates that heterogeneity materially **restructures the social dynamics of discovery**.
> >
> > We have also included an annotated trajectory of a single agent from the `ASCollab` system (`Fig 9`), demonstrating how exposure to collaboration and shared memory changes its research trajectory—from simple biomarkers, to pathway-level analyses, to system-level integrative modeling—mirroring the way real scientists learn from their communities.
> >
> > These new results offer empirical evidence that **heterogeneity, dynamic collaboration, and shared social memory are important drivers of improvements in novelty, quality, and diversity**. While not a theoretical analysis, we hope this empirical analysis clearly highlights the role that each social component plays in encouraging cumulative and open-ended discovery.
> >
> >
> > ---
> >
> > *Thank you again for your careful evaluation, which helped us strengthen our work. We hope our responses and new results fully addressed your concerns; we would be glad to clarify or expand on any remaining points.*

---

### Official Review · Reviewer_zfHW · 2025-10-27

**Soundness:** 2
**Presentation:** 2
**Contribution:** 2
**Rating:** 4
**Confidence:** 3

**Summary:**

The paper introduces the task of hypothesis hunting and presents AScience, a framework that treats scientific discovery as an interaction among agents, their networks, and shared evaluation norms. The authors implement this framework as ASCollab, a distributed system of LLM based research agents with diverse behaviors. These agents organize themselves into evolving collaboration networks and continuously produce findings while peer reviewing one another under common evaluation standards.

**Strengths:**

* The overall framework is fairly comprehensive and logically coherent.
* Includes concrete downstream tasks.

**Weaknesses:**

* The novelty of paper is limited; it essentially applies a multi-agent approach in the biology domain. What it calls “hypothesis hunting” is, in my view, simply scientific discovery.
* The experimental evaluation is insufficient; see the Questions section.

**Questions:**

* How does the number of agents affect performance? I suggest scaling the number of agents.
* How are other backbones used in the system, such as Qwen or LLaMA?
* I believe ablation studies removing certain types of agents are missing; for example, what are the results if the review stage is omitted?
* Please explain exactly what the difference is between scientific discovery and hypothesis hunting.

---

> ### Author Response · Authors · 2025-11-25
> **Response to Reviewer zfHW (Part 1/3)**
>
> *Thank you for taking the time to review our work and for providing clear, constructive feedback. We appreciate your recognition of the framework’s coherence and its concrete downstream applications. Below, we address your key concerns.*
>
> ---
>
> ### [P1] Clarification of Novelty
>
> We appreciate the reviewer’s thoughtful comments. While our system employs multi-agent methods and is demonstrated in the biological domain, we see our primary contribution as a conceptual premise: that **scientific discovery should be modeled as an emergent social process**, and we formalize this idea through three interacting components:
> 1. **Distributed epistemic heterogeneity**: agents differ in research style, depth, and exploratory behavior.
> 2. **Endogenous collaboration networks**: connections evolve dynamically through reputation, attention, and mutual relevance, and not fixed workflows.
> 3. **Shared social memory and internal evaluation norms**: peer review and meta-review surface promising ideas and support cumulative knowledge building in the absence of predefined tasks or ground truth.
>
> In this light, the system implementation `ASCollab` is one concrete instantiation of such concepts, designed to illustrate how these components jointly enable open-ended scientific exploration.
>
> **Contrast with prior works.** Most existing multi-agent scientific systems are fundamentally **task-driven**: agents collaborate (e.g., AI Co-Scientist) or compete (AI Scientist) to answer a specific user-defined research question. Their social dynamics are therefore geared toward converging on a single optimal result. Our framework, `AScience`, is designed for a different setting, one characterized by **divergent exploration, cumulative refinement, and the maintenance of a diverse frontier of high-quality, novel hypotheses** (which we further elaborate on in **[P3]**). This leads to essential technical differences:
>
> |  | Collaborative systems | Competitive systems | Social system (ours) |
> |---|---|---|---|
> | Epistemic outcome | Convergent: single solution | Convergent: fittest solution | Divergent: diversity–quality–novelty frontier |
> | Goal structure | Shared global objective | Competing / adversarial objectives | No centralized objective; open-ended exploration |
> | Agent heterogeneity | Functional roles | Fitness differences | Epistemic: distinct research behaviors |
> | Interaction pattern | Mostly collaborative | Mostly competitive | Dynamic mixture driven by attention, reputation, evolving expertise |
> | Network evolution | Fixed workflow: static hierarchy, team roles | Fixed interaction: static population structure | Endogenously evolving collaboration and attention networks |
> | Memory | Shared task memory | Private memory | Global archive (social memory) + private memories |
> | Examples | AI Co-scientist, ChemCrow, swarm robotics | AI Scientist, AlphaGo, evolutionary algorithms | ASCollab, human scientific ecosystem |
>
> **Empirical validation of this distinction.** As we expand on in **[P2]**, our ablation studies directly test the contribution of each component of our social system. These findings affirm that the improvements arise from the social dynamics our framework formalizes. We will revise the manuscript to make these distinctions clearer and more prominently articulated in the Introduction.
>
> ---
>
> ### [P2] Additional Ablation Studies
>
> Given our previous framing, we believe the best way to isolate the effect of *social dynamics* is to hold the underlying LLM, tools, and research environment fixed and remove only the social layer. This motivated our existing baseline with a population of independent agents. To further clarify the contributions of each social component, we have added an expanded ablation suite:
>
> | Ablation | Collaborations | Heterogeneous | # agents / distributed |
> |---|---|---|---|
> | single-agent (GPT-5) | N/A | N/A | 1 / ❌ |
> | independent_homogeneous | ❌ | ❌ | 16 / ✔️ |
> | collaborating_homogeneous | ✔️ | ❌ | 16 / ✔️ |
> | ASCollab | ✔️ | ✔️ | 16 / ✔️ |
>
> Notably, heterogeneity in the `ASCollab` setting (our full system) now comes from both different LLM backbones (we integrate GPT-4o, GPT-5, DeepSeek-3.1, Llama-3.3) and distinct agent-specific epistemic profiles.
>
> **Key takeaways.** We refer you to `App A1`, which presents detailed analyses. Here, we describe the key findings in brief:
> 1. **ASCollab produces the strongest hypotheses in direct head-to-head tournaments.** On KIRC, human experts compared the n-th best paper from each ablation in a four-way tournament (thus giving n=25 tournaments). Across all tournaments, ASCollab’s submissions were judged most scientifically significant, outperforming both `collab_homo` and the single powerful GPT-5 agent (`Fig 6a`).

---

> > ### Author Response · Authors · 2025-11-25
> > **Response to Reviewer zfHW (Part 2/3)**
> >
> > 2. **Collaboration (even without heterogeneity) substantially improves quality-novelty.** Compared to `ind_homo`, the `collab_homo` condition produced higher-quality and higher-novelty findings, reflecting the value of endogenous collaboration in consolidating high-value findings, and surfacing promising regions for further exploration.
> > 3. **Epistemic heterogeneity is a driver of novelty and diversity.** Comparing `collab_homo` vs. `ASCollab` (collaborative but heterogeneous agents), the heterogeneous population produced hypotheses that were more novel, conceptually broader, and explored more distinct research directions (`Fig 6c`).
> > 4. **A strong single agent is competitive—but fundamentally limited.** While the GPT-5 single agent generated high-quality hypotheses, its findings were much less diverse, clustering around a narrower conceptual space (as shown in `Fig 6b`). Without distributed and collaborative exploration, it cannot cover the novelty–quality–diversity frontier reached by the collective system.
> >
> > In addition to the findings above (which were regarding scientific outcomes produced by the system), we also analyzed how heterogeneity and collaboration affects individual agent behavior and network evolution.
> > * Our **agent-behavior analysis** (`Fig 7`) shows that heterogeneous agents develop **distinct methodological niches**, differ widely in code generation and tool use, and pursue different exploration–exploitation strategies. (In contrast, homogeneous agents behave nearly identically.)
> > * **Network-evolution analysis** (`Fig 8`) reveals that ASCollab exhibits higher connectivity, more cross-cluster interaction, lower influence inequality, and greater cumulative exchange of ideas compared to homogeneous collaborators. This demonstrates that heterogeneity materially **restructures the social dynamics of discovery**.
> >
> > We have also included an annotated trajectory of a single agent from the `ASCollab` system (`Fig 9`), demonstrating how exposure to collaboration and shared memory changes its research trajectory—from simple biomarkers, to pathway-level analyses, to system-level integrative modeling—mirroring the way real scientists learn from their communities.
> >
> > These new results offer direct, empirical evidence that **heterogeneity, dynamic collaboration, and shared social memory (rather than agent count or raw model strength) are important drivers of improvements in novelty, quality, and diversity**. We hope this clarifies the conceptual novelty of our approach and its distinction from prior multi-agent research systems.
> >
> > ---
> >
> > ### [P3] Distinction between "Hypothesis Hunting" and "Scientific Discovery"
> >
> > We appreciate the opportunity to clarify this point. While hypothesis hunting is indeed a sub-process within the broader umbrella of scientific discovery, the term “scientific discovery” is too broad to serve as an actionable technical problem. It spans many stages: hypothesis formation, experimental design, data collection, and wet-lab validation, most of which lie outside the scope of our work.
> >
> > The closest existing sub-process is **hypothesis generation**, and it is precisely in contrast to this literature (e.g., AI Scientist, AI Co-Scientist) that we define **hypothesis hunting**. As detailed in `Sec 2.2`, hypothesis hunting is distinguished by three structural properties:
> >
> > 1. **Open-ended exploration**: Standard hypothesis-generation systems operate in a *task-driven* regime, producing hypotheses in response to a predefined prompt or question. In contrast, hypothesis hunting requires agents to *identify the research questions themselves* through divergent and meaningful exploration of large-scale scientific datasets.
> > 2. **Cumulative and continuous**: The term "hunting" reflects that discovery is long-horizon and iterative: hypotheses must be refined, contextualized, and integrated into a shared, evolving research landscape. This necessitates shared memory and persistent accumulation, unlike task-based research, which ends once a predefined goal is achieved.
> > 3. **Datasets-defined hypothesis space and endogenous evaluation**: In hypothesis hunting, the datasets *define the hypothesis space*, and because no predefined task or ground-truth target exists, evaluation must rely on internal mechanisms: peer feedback, socially defined norms, and attention/reputation dynamics to adjudicate significance and novelty.
> >
> > In our view, these structural differences—open-ended exploration, cumulative knowledge building, and endogenous evaluation—justify distinguishing hypothesis hunting from conventional hypothesis generation. To avoid ambiguity, we will revise the introduction to explicitly relate the two terms and clarify the motivation for defining this new problem setting.
> >
> > Please let us know if our explanation clarifies the differences.

---

> > > ### Author Response · Authors · 2025-11-25
> > > **Response to Reviewer zfHW (Part 3/3)**
> > >
> > > ### [P4] Network Scaling
> > >
> > > Thank you for raising this important point. We fully agree that understanding how social dynamics behave at much larger scales is an exciting direction. At present, however, scaling to 100+ agents poses a practical engineering constraint: each agent requires an isolated, persistent execution sandbox, and our academic compute environment cannot yet support such massively parallel deployments.
> > >
> > > We are actively exploring more scalable infrastructure to enable larger-population experiments. In the meantime, our ablation studies (summarized in **[P2]**) isolate the contribution of each social component to observed improvements. In our view, **scale changes the magnitude of the gains, not the underlying mechanism**. We will report large-scale results when/if they become feasible.
> > >
> > > ---
> > >
> > > ### [P5] Integration of Diverse LLM Backbones
> > >
> > > We appreciate the reviewer’s questions regarding different LLM backbones. Our new ablation experiments (detailed in **[P2]** and `App A1`) directly address this concern:
> > >
> > > **Heterogeneous LLM Backends.** We instantiated ASCollab with a mixed population of agents powered by GPT-5, GPT-4o, DeepSeek-3.1, and Llama-3.3. The heterogeneous-backbone `ASCollab` achieved the strongest performance on the novelty–quality–diversity frontier across all runs. **Model diversity amplified epistemic heterogeneity**, leading to broader exploration and richer collaboration patterns (illustrated through the analyses in `Fig 7`).
> > >
> > > **Takeaway.** This demonstrates that the framework is model-agnostic and can benefit from mixing LLMs (to further boost heterogeneity) rather than relying on a single powerful backbone.
> > >
> > > ---
> > >
> > > ### [P6] Evaluation on Additional Domains
> > >
> > > Additionally, we wanted to demonstrate generality beyond cancer genomics. During the rebuttal period, we deployed `ASCollab` on two additional, scientifically distinct datasets, without modifying the architecture or social mechanisms:
> > >
> > > 1. **NHANES (Public Health)**: harmonized participant-level dataset (with demographic, clinical, lifestyle, exposure, clinical variables and myocardial-infarction outcome) commonly used in epidemiology research.
> > > 2. **MYC-perturbation multi-omics dataset (Proprietary)**: A specialized dataset integrating eNET-seq RNAPII occupancy, bulk RNA-seq expression, and TRRUST TF–target interactions under MYC activation/inhibition. This represents a somewhat niche, high-dimensional regulatory dataset not commonly encountered in LLM training corpora.
> > >
> > > **Results:** We presented detailed analysis in `App A2`. In both domains, the system successfully adapted its tool usage and research behaviors without architectural changes, generated several domain-plausible and non-obvious findings:
> > > * **NHANES — Acidosis × Insulin-Resistance Interaction**: ASCollab identified a synergistic interaction between low serum bicarbonate and high TyG index in elevating MI risk, with serum albumin moderating this effect, surfacing a cross-metabolic interaction not typically examined in standard epidemiologic workflows.
> > > * **NHANES — Liver–Kidney Metabolic Flexibility Marker:** The system proposed the AST/Creatinine ratio as a protective marker of metabolic flexibility with a non-linear effect, particularly in metabolic-syndrome subgroups, highlighting a physiologically interpretable but underutilized biomarker.
> > > * **MYC-Perturbation Dataset — MYC-Aligned Anabolic Program:** ASCollab identified a coherent regulatory triad (EEF2, SLC7A5, FASN) that together constitute a MYC-driven anabolic program, integrating transcriptional, pausing, and pathway data into a unified mechanistic model.
> > >
> > > **Takeaway.** When considered along with existing results, we have demonstrated `ASCollab` operating over seven distinct modalities: RNA-Seq, proteomics, pathway/topology data, drugs & perturbations databases, Net-Seq, and population-level epidemiology. This breadth highlights that the system is **modality-agnostic and its social architecture transfers effectively to new scientific regimes**.
> > >
> > > **Note.** Due to limited expert availability during the rebuttal period, we provide qualitative domain-expert assessments rather than full rubric-based scoring. Nevertheless, in both datasets, ASCollab produced novel, high-quality, and domain-plausible hypotheses. We will include expanded expert evaluations in the camera-ready version.
> > >
> > > ---
> > >
> > > *Thank you again for your thoughtful evaluation. We hope our responses and newly added results help resolve your concerns, and we would be glad to elaborate further on any remaining points. We appreciate your time and consideration.*

---

### Official Review · Reviewer_pLNm · 2025-10-28

**Soundness:** 3
**Presentation:** 2
**Contribution:** 3
**Rating:** 6
**Confidence:** 4

**Summary:**

This paper, submitted to ICLR 2026, addresses the challenge of exploratory discovery in large-scale scientific datasets (e.g., cancer genomics, biobanks) through a framework called AScience and its instantiation ASCollab. The core concept, termed "hypothesis hunting," refers to the continuous, diverse exploration of large datasets to surface promising findings for human validation—addressing limitations of human-led research (scale and coordination).

AScience models scientific progress as a dynamic system with four components: an epistemic landscape of research approaches, heterogeneous scientific agents, attention-routing networks, and shared evaluation norms. ASCllab, a distributed system built on this framework, uses LLM-based agents (with heterogeneous expertise and behaviors) that self-organize into evolving networks. These agents generate findings, collaborate, and undergo a structured peer-review process (review + meta-review) to curate high-quality results into a shared archive.

Empirical evaluations on three TCGA cancer cohorts (KIRC, PAAD, DLBC) show that ASCllab outperforms independent agents: its findings are more novel, higher-quality, and diverse, including rediscoveries of established biomarkers (e.g., BIRC5 in KIRC), extensions of known pathways (e.g., ferroptosis), and proposals of new therapeutic targets (e.g., SLC5A2 in PAAD). The paper concludes that socially structured agent networks enable scalable, cumulative hypothesis hunting, though wet-lab validation remains necessary.

**Strengths:**

(1) The writing is easy to follow.

(2) This paper fills a Critical Gap. Unlike existing autonomous science systems (e.g., AI Scientist, AI Co-Scientist) that focus on answering predefined research questions, this work formalizes "hypothesis hunting" as a distinct, open-ended problem setting. It explicitly addresses the need for cumulative exploration (not just goal-driven convergence) and models scientific progress as a social process—mirroring human scientific communities (collaboration, peer review, knowledge accumulation).

**Weaknesses:**

(1) Incomplete experimental validation. All experiments focus on cancer genomics (TCGA). The paper does not test ASCllab on other large-scale datasets (e.g., Earth reanalyses, cell atlases) mentioned in the introduction. It is unclear if the framework’s social dynamics (e.g., collaboration, peer review) would translate to fields with different evaluation norms (e.g., physics, climate science) or data types (e.g., image-based cell atlases vs. tabular genomics).

(2) The experiments use only 16 agents over 40 rounds. While this suffices to demonstrate proof of concept, it is unclear how the system would behave with larger populations (e.g., 100+ agents) or longer timeframes. For example:
Would network dynamics become unmanageable (e.g., information overload in the archive)?
Would agent diversity persist, or would specialization converge over time?


(3) Expert Assessment is somewhat subjective: While expert evaluation is a strength, the paper relies on a single domain expert for KIRC (and two for case studies). There is no inter-rater reliability analysis—different experts might score novelty/quality differently, especially for "borderline" findings (e.g., N3/N4 novelty). Are there any cross-validation or alogrithm-based evaluation.


(4) The meta-review process uses "tournament-style" relative scoring of clustered submissions, but the paper does not explain how clusters are defined (e.g., thematic similarity metrics) or how meta-reviewers calibrate scores across clusters. This could introduce bias if clusters are unevenly sized or themed.

**Questions:**

Please see the weakness.

---

> ### Author Response · Authors · 2025-11-25
> **Response to Reviewer  pLNm (Part 1/3)**
>
> *Thank you for your constructive review. We are encouraged by your recognition that our work addresses a critical gap by modeling scientific progress as a social process. Below, we address your four key concerns.*
>
> ---
>
> ### [P1] Evaluation on Additional Domains
>
> We agree that demonstrating generality beyond cancer genomics is essential. During the rebuttal period, we deployed `ASCollab` on two additional, scientifically distinct datasets, without modifying the architecture or social mechanisms:
>
> 1. **NHANES (Public Health)**: harmonized participant-level dataset (with demographic, clinical, lifestyle, exposure, clinical variables, and myocardial-infarction outcome) commonly used in epidemiology research.
> 2. **MYC-perturbation multi-omics dataset (Proprietary)**: A specialized dataset integrating eNET-seq RNAPII occupancy, bulk RNA-seq expression, and TRRUST TF–target interactions under MYC activation/inhibition. This represents a somewhat niche, high-dimensional regulatory dataset not commonly encountered in LLM training corpora.
>
> **Results:** We presented detailed analysis in `App A2`. In both domains, the system successfully adapted its tool usage and research behaviors without architectural changes, generated several domain-plausible and non-obvious findings:
> * **NHANES — Acidosis × Insulin-Resistance Interaction**: ASCollab identified a synergistic interaction between low serum bicarbonate and high TyG index in elevating MI risk, with serum albumin moderating this effect, surfacing a cross-metabolic interaction not typically examined in standard epidemiologic workflows.
> * **NHANES — Liver–Kidney Metabolic Flexibility Marker:** The system proposed the AST/Creatinine ratio as a protective marker of metabolic flexibility with a non-linear effect, particularly in metabolic-syndrome subgroups, highlighting a physiologically interpretable but underutilized biomarker.
> * **MYC-Perturbation Dataset — MYC-Aligned Anabolic Program:** ASCollab identified a coherent regulatory triad (EEF2, SLC7A5, FASN) that together constitute a MYC-driven anabolic program, integrating transcriptional, pausing, and pathway data into a unified mechanistic model.
>
> **Takeaway.** When considered along with existing results, we have demonstrated `ASCollab` operating over seven distinct modalities: RNA-Seq, proteomics, pathway/topology data, drugs & perturbations databases, Net-Seq, and population-level epidemiology. This breadth highlights that the system is **modality-agnostic, and its social architecture transfers effectively to new scientific regimes**.
>
> **Note #1.** Due to limited expert availability during the rebuttal period, we provide qualitative domain-expert assessments rather than full rubric-based scoring. Nevertheless, in both datasets, ASCollab produced novel, high-quality, and domain-plausible hypotheses. We will include expanded expert evaluations in the camera-ready version.
>
> **Note #2.** Our choice of additional evaluations is constrained by domain expertise, and we acknowledge that `ASCollab` has not yet been systematically tested in non-biological domains. While such studies are difficult to conduct within the rebuttal period, we now explicitly highlight this as an important direction for future work. We remain confident that the framework’s core principles are broadly domain-agnostic.
>
> ---
>
> ### [P2] Generality of Social Dynamics and Norms
>
> We appreciate this insightful question. We agree that different scientific fields vary in their evaluation norms and data modalities, and we address the concern in two parts.
>
> **(1) Social dynamics are domain-general abstractions.**
> The social dynamics in `AScience`, including evolving collaboration networks, heterogeneous agents, and cumulative shared memory, are **domain-general abstractions**, inspired by how scientific communities operate across disciplines. These mechanisms do not assume a particular data type or scientific workflow, which can be swapped out without altering the social aspects of the process.
>
> As described in **[P1]**, our deployment of ASCollab on NHANES and Myc-perturbation data confirms that these mechanisms successfully route attention and accumulate knowledge even when the underlying data types and domain concepts differ radically from genomics.
>
> **(2) Framework generality vs. mechanism instantiation.**
> We distinguish between `AScience` (the general formalism) and `ASCollab` (our specific instantiation). While the necessity of social dynamics is central to our thesis, the specific mechanisms can be designed to reflect domain norms (e.g., open vs. closed review, varying reproducibility norms, and incentives for cross-disciplinary collaboration). We do not believe that the concrete **mechanism designs** (in ASCollab) are universally optimal (across scientific fields); rather, it serves as a vehicle for us to highlight that **some form of social structure is essential for cumulative and open-ended discovery**.

---

> > ### Author Response · Authors · 2025-11-25
> > **Response to Reviewer pLNm (Part 2/3)**
> >
> > To this end, our new ablation studies (in **[P3]**) further show that these social components are what sustain divergent exploration along the quality–novelty–diversity frontier. We have revised the Discussion to clearly separate the generalizable social framework from domain-specific mechanism choices, and to outline how agentic research communities may develop norms distinct from human scientific fields.
> >
> > ---
> >
> > ### [P3] Network Behavior with Scaling
> >
> > Thank you for raising this interesting point.
> >
> > **Practical vs. conceptual limitations.** Scaling beyond 16 agents requires substantial infrastructure, as each agent maintains its own persistent sandbox with tool access, code execution, and private memory, making very large populations difficult to run within the compute infrastructure available to our academic lab. We emphasize, however, that this is a **practical limitation** rather than a conceptual one; regarding your specific comments:
> > - **Information overload**: agents do not process the entire archive or private memory linearly. All agent access occurs through RAG-style retrieval, decoupling network size (and archive size) from the agent's context window, and is thus localized.
> > - **Specialization and convergence**: As shown in `Figs 4–5`, agents maintain stable but distinct epistemic behaviors over time (e.g., exploratory vs. exploitative trajectories), and collaboration patterns remain diverse, suggesting that collapse to a single specialization is unlikely.
> >
> > **Additional ablations.** Because our central thesis concerns the **necessity of social mechanisms**, and our compute budget limits large-population runs, we focus on ablating these components rather than scaling agent count. In our view, **scale changes the magnitude of the gains, not the underlying mechanism**. The new ablations isolate the contribution of (1) collaboration, (2) epistemic heterogeneity, (3) shared social memory, and (4) distributed agent populations:
> >
> > | Ablation | Collaborations | Heterogeneous | Social memory | # agents / distributed |
> > |---|---|---|---|---|
> > | single_agent (GPT-5) | N/A | N/A | ❌ | 1 / ❌ |
> > | independent_homogeneous | ❌ | ❌ | ❌ | 16 / ✔️ |
> > | collaborating_homogeneous | ✔️ | ❌ | ✔️ | 16 / ✔️ |
> > | collaborating_heterogeneous (ASCollab) | ✔️ | ✔️ | ✔️ | 16 / ✔️ |
> >
> > **Key takeaways.** We refer you to `App A1`, which presents detailed analyses. Here, we describe the key findings in brief:
> > 1. **ASCollab produces the strongest hypotheses in direct head-to-head tournaments.** On KIRC, human experts compared the n-th best paper from each ablation in a four-way tournament (thus giving n=25 tournaments). Across all tournaments, ASCollab’s submissions were judged most scientifically significant, outperforming both `collab_homo` and the single powerful GPT-5 agent (`Fig 6a`).
> > 2. **Collaboration (even without heterogeneity) substantially improves quality-novelty.** Compared to `ind_homo`, the `collab_homo` condition produced higher-quality and higher-novelty findings, reflecting the value of endogenous collaboration in consolidating high-value findings, and surfacing promising regions for further exploration.
> > 3. **Epistemic heterogeneity is a driver of novelty and diversity.** Comparing `collab_homo` vs. `ASCollab` (collaborative but heterogeneous agents), the heterogeneous population produced hypotheses that were more novel, conceptually broader, and explored more distinct research directions (`Fig 6c`).
> > 4. **A strong single agent is competitive—but fundamentally limited.** While the GPT-5 single agent generated high-quality hypotheses, its findings were much less diverse, clustering around a narrower conceptual space (as shown in `Fig 6b`). Without distributed and collaborative exploration, it cannot cover the novelty–quality–diversity frontier reached by the collective system.
> >
> > In addition to the findings above (which were regarding scientific outcomes produced by the system), we also analyzed how heterogeneity and collaboration affect individual agent behavior and network evolution.
> > * Our **agent-behavior analysis** (`Fig 7`) shows that heterogeneous agents develop **distinct methodological niches**, differ widely in code generation and tool use, and pursue different exploration–exploitation strategies. (In contrast, homogeneous agents behave nearly identically.)
> > * **Network-evolution analysis** (`Fig 8`) reveals that ASCollab exhibits higher connectivity, more cross-cluster interaction, lower influence inequality, and greater cumulative exchange of ideas compared to homogeneous collaborators. This demonstrates that heterogeneity materially **restructures the social dynamics of discovery**.

---

> > > ### Author Response · Authors · 2025-11-25
> > > **Response to Reviewer pLNm (Part 3/3)**
> > >
> > > We have also included an annotated trajectory of a single agent from the `ASCollab` system (`Fig 9`), demonstrating how exposure to collaboration and shared memory changes its research trajectory—from simple biomarkers, to pathway-level analyses, to system-level integrative modeling—mirroring the way real scientists learn from their communities.
> > >
> > > **Takeaway.** These results underscore that the components underpinning our social process of discovery functions robustly in moderate-size populations and provide a strong foundation for future work exploring larger-scale deployments.
> > >
> > >
> > > ---
> > >
> > > ### [P4] Reliability of Expert Analysis
> > >
> > > Thank you for raising this important point. Evaluating open-ended scientific hypotheses is **inherently challenging**: unlike other ML tasks, there is no benchmark or automated metric for assessing the novelty, quality, or biological plausibility of a hypothesis. As is standard in hypothesis-generation settings, our evaluation therefore relies on domain experts.
> > >
> > > **Mitigating subjectivity.** We agree that expert judgment introduces variance, and we addressed this by using an anchor-based rubric (`Table 2`) rather than open-ended scoring. This reduces drift in novelty and quality assessments, especially for borderline cases (e.g., N3/N4).
> > >
> > > **Cross-domain robustness.** Although the rebuttal timeline prevented a full inter-rater reliability study for KIRC, our evaluations on two additional datasets (NHANES and a MYC-perturbation dataset) required involvement of two additional, independent domain experts. Both assessed `ASCollab`’s outputs as scientifically plausible and potentially significant in their fields. This suggests that the system’s ability to generate credible hypotheses is **not tied to a single expert and generalizes across domains and data modalities**.
> > >
> > > **Agreement with external literature.** We also emphasize that several high-scoring hypotheses were later independently supported or partially validated by newly published scientific studies. This post-hoc agreement indicates that expert ratings were not arbitrary and **meaningfully tracked real scientific value as judged by the broader community**.
> > >
> > > We will update the Discussion to clarify these points and to outline plans for broader expert evaluation and reliability analysis in future work.
> > >
> > > ---
> > >
> > >
> > > ### [P5] Clarification on Meta-review Process
> > >
> > > We clarify that submissions are clustered using embeddings of the title and abstract, such that each tournament (1) consists of thematically similar submissions and (2) is uniform in size (via constrained K-means algorithm).
> > >
> > > This approach allows the meta-reviewer to directly contrast submissions addressing similar topics, where acceptance is determined by relative ranking within each tournament (i.e., the "winner" of the tournament gets accepted). This reduces drift from cross-tournament scoring variance by the meta-reviewing agents. Additionally, across rounds, the meta-reviewing agent has access to its past meta-reviews, providing temporal consistency in ranking behavior.
> > >
> > > ---
> > >
> > > *Thank you again for your careful evaluation, which helped us strengthen our work. We hope our responses and new results fully addressed your concerns; we would be glad to clarify or expand on any remaining points.*

---

### Official Review · Reviewer_KDz9 · 2025-10-31

**Soundness:** 3
**Presentation:** 3
**Contribution:** 2
**Rating:** 2
**Confidence:** 5

**Summary:**

The paper presents AScience, a multi-agent framework for “hypothesis hunting,” i.e., the autonomous exploration of large-scale scientific datasets to surface potential insights. Each agent functions as a simulated researcher that can analyze data, generate hypotheses, peer-review others’ work, and exchange findings within an evolving social network. The authors implement this system as ASCollab and evaluate it on cancer genomics datasets (TCGA), showing that such socially structured agent networks can produce diverse, novel, and scientifically plausible hypotheses, some of which rediscover known biomarkers or propose new therapeutic leads. The approach is conceptually related to prior multi-agent simulation efforts (e.g., Simulacra, SocialLab, etc.) but extends them to emulate a full research community interacting around a dataset.

**Strengths:**

The vision is compelling. The paper introduces an ambitious and creative idea: deploying a network of autonomous agents to “hunt” for hypotheses within complex scientific datasets. Conceptually, this aligns with the broader movement toward AI-for-Science systems that go beyond automation to exploration and reasoning. The idea of coupling such a system with downstream validation agents (e.g., experimental design, simulation, or lab integration frameworks like Biomni) could, in principle, enable a closed loop of scientific discovery. While the current implementation is early, the long-term vision, AI agents collaborating, critiquing, and refining hypotheses like a synthetic scientific community, is inspiring and potentially high-impact.

**Weaknesses:**

The use of the term “hypothesis hunting” is confusing and somewhat unnecessary. The established term in the literature is hypothesis generation, and it is unclear whether the authors intend a substantive distinction or simply rebrand existing ideas. Framing aside, multi-agent hypothesis generation has already been explored in prior works cited in the related-work section. However, none of these prior systems are directly benchmarked here, the only comparison is to independent single-agent runs, making it difficult to assess what is actually improved by the proposed approach.

Methodologically, the system relies on standard multi-agent orchestration built on top of GPT-4o, a model that has since been surpassed by more capable successors (e.g., GPT-5/5-Pro). This raises an important question: would a stronger single model already outperform or render unnecessary the proposed multi-agent framework for hypothesis generation? Without such comparisons, it is difficult to isolate the contribution of the agentic coordination itself. In addition, all experiments are confined to a single dataset (TCGA), despite the introduction referencing a broader range of large-scale scientific datasets. Evaluating across multiple LLMs and diverse domains would substantially strengthen both the generality and the empirical rigor of the paper.

Overall, while the engineering is solid, the paper lacks clear methodological or conceptual novelty. If the contribution lies in a superior implementation of multi-agent hypothesis generation, this should be validated through head-to-head benchmarks against prior systems. In its current form, the paper feels more like an incremental composition of existing ideas rather than a new learning or inference principle.

**Questions:**

See Weaknesses

---

> ### Author Response · Authors · 2025-11-25
> **Response to Reviewer KDz9 (Part 1/3)**
>
> *We appreciate your thoughtful and constructive review. We are encouraged that our framing of scientific progress as a social process resonates with you, and we thank you for engaging deeply with the work. Below, we address each of your key concerns.*
>
> ---
>
> ### [P1] Distinction between "Hypothesis Hunting" and "Hypothesis Generation"
>
>
> Thank you for the opportunity to clarify this terminology. Our intention is not to rebrand hypothesis generation, but to formalize a more specific problem setting with requirements that differ from the task-based discovery commonly studied. As described in `Section 2.2`, hypothesis hunting is defined by three requirements:
>
> 1. **Open-ended exploration**: Standard hypothesis-generation systems operate in a *task-driven* regime, producing hypotheses in response to a predefined prompt or question. In contrast, hypothesis hunting requires agents to *identify the research questions themselves* through divergent and meaningful exploration of large-scale scientific datasets.
> 2. **Cumulative and continuous**: The term "hunting" reflects that discovery is long-horizon and iterative: hypotheses must be refined, contextualized, and integrated into a shared, evolving research landscape. This necessitates shared memory and persistent accumulation, unlike task-based research, which ends once a predefined goal is achieved.
> 3. **Datasets-defined hypothesis space and endogeneous evaluation**: In hypothesis hunting, the datasets *define the hypothesis space*, and because no predefined task or ground-truth target exists, evaluation must rely on internal mechanisms: peer feedback, socially defined consensus, and attention/reputation dynamics to adjudicate significance and novelty.
>
> In our view, these structural differences—open-ended exploration, cumulative knowledge building, and endogenous evaluation—justify distinguishing hypothesis hunting from conventional hypothesis generation. To avoid ambiguity, we will revise the introduction to explicitly relate the two terms and clarify the motivation for defining this new problem setting.
>
> ---
>
> ### [P2] Clarification of Novelty
>
> Thank you for raising this point. We address the concern in three parts.
>
> **Baselines.** Only a small subset of prior multi-agent systems directly target hypothesis generation; most focus on automating scientific workflows. Among the closest systems, AI Scientist is specialized for machine-learning research pipelines and cannot be meaningfully ported to biological hypothesis discovery without substantial re-engineering, while AI Co-Scientist is closed-source. Neither therefore serves as a feasible plug-and-play baseline for our setting.
>
> **Contributions.** From our perspective, our primary contribution is not a new multi-agent protocol/architecture, but a *simple premise*: **discovery should be modeled as an emergent social process**, which we formalized through three interacting components
> 1. **Distributed epistemic heterogeneity**: agents differ in research style, depth, and exploratory behavior.
> 2. **Endogenous collaboration networks**: connections evolve dynamically through reputation, attention, and mutual relevance, and not fixed workflows.
> 3. **Shared social memory and internal evaluation norms**: peer review and meta-review surface promising ideas and support cumulative knowledge building in the absence of predefined tasks or ground truth.
>
> In this light, the system implementation `ASCollab` is one concrete instantiation of such concepts, designed to illustrate how these components jointly enable open-ended scientific exploration.
>
> **Additional analyses.** Given this framing, we believe the best way to isolate the effect of “social dynamics” is to hold the underlying LLM, tools, and research environment fixed and remove only the social layer. This motivated our existing baseline with a population of independent agents. To further clarify the contributions of each social component, we have added an expanded ablation suite:
>
> | Ablation | Collaborations | Heterogeneous | # agents / distributed |
> |---|---|---|---|
> | single-agent (GPT-5) | N/A | N/A | 1 / ❌ |
> | independent_homogeneous | ❌ | ❌ | 16 / ✔️ |
> | collaborating_homogeneous | ✔️ | ❌ | 16 / ✔️ |
> | ASCollab | ✔️ | ✔️ | 16 / ✔️ |
>
> Notably, heterogeneity in the `ASCollab` setting (our full system) now comes from both different LLM backbones (we integrate GPT-4o, GPT-5, DeepSeek-3.1, Llama-3.3) and distinct agent-specific epistemic profiles.

---

> > ### Author Response · Authors · 2025-11-25
> > **Response to Reviewer KDz9 (Part 2/3)**
> >
> > **Key takeaways.** We refer you to `App A1`, which presents detailed analyses. Here, we describe the key findings in brief:
> > 1. **ASCollab produces the strongest hypotheses in direct head-to-head tournaments.** On KIRC, human experts compared the n-th best paper from each ablation in a four-way tournament (thus giving n=25 tournaments). Across all tournaments, ASCollab’s submissions were judged most scientifically significant, outperforming both `collab_homo` and the single powerful GPT-5 agent (`Fig 6a`).
> > 2. **Collaboration (even without heterogeneity) substantially improves quality-novelty.** Compared to `ind_homo`, the `collab_homo` condition produced higher-quality and higher-novelty findings, reflecting the value of endogenous collaboration in consolidating high-value findings, and surfacing promising regions for further exploration.
> > 3. **Epistemic heterogeneity is a driver of novelty and diversity.** Comparing `collab_homo` vs. `ASCollab` (collaborative but heterogeneous agents), the heterogeneous population produced hypotheses that were more novel, conceptually broader, and explored more distinct research directions (`Fig 6c`).
> > 4. **A strong single agent is competitive—but fundamentally limited.** While the GPT-5 single agent generated high-quality hypotheses, its findings were much less diverse, clustering around a narrower conceptual space (as shown in `Fig 6b`). Without distributed and collaborative exploration, it cannot cover the novelty–quality–diversity frontier reached by the collective system.
> >
> > In addition to the findings above (which were regarding scientific outcomes produced by the system), we also analyzed how heterogeneity and collaboration affect individual agent behavior and network evolution.
> > * Our **agent-behavior analysis** (`Fig 7`) shows that heterogeneous agents develop **distinct methodological niches**, differ widely in code generation and tool use, and pursue different exploration–exploitation strategies. (In contrast, homogeneous agents behave nearly identically.)
> > * **Network-evolution analysis** (`Fig 8`) reveals that ASCollab exhibits higher connectivity, more cross-cluster interaction, lower influence inequality, and greater cumulative exchange of ideas compared to homogeneous collaborators. This demonstrates that heterogeneity materially **restructures the social dynamics of discovery**.
> >
> > We have also included an annotated trajectory of a single agent from the `ASCollab` system (`Fig 9`), demonstrating how exposure to collaboration and shared memory changes its research trajectory—from simple biomarkers, to pathway-level analyses, to system-level integrative modeling—mirroring the way real scientists learn from their communities.
> >
> > These new results offer direct, empirical evidence that **heterogeneity, dynamic collaboration, and shared social memory (rather than agent count or raw model strength) are important drivers of improvements in novelty, quality, and diversity**. We hope this clarifies the conceptual novelty of our approach and its distinction from prior multi-agent research systems.
> >
> > ---
> >
> > ### [P3] Generality: Evaluation on Additional Domains
> >
> > We agree that demonstrating generality beyond cancer genomics is essential. During the rebuttal period, we deployed `ASCollab` on two additional, scientifically distinct datasets, without modifying the architecture or social mechanisms:
> >
> > 1. **NHANES (Public Health)**: harmonized participant-level dataset (with demographic, clinical, lifestyle, exposure, clinical variables and myocardial-infarction outcome) commonly used in epidemiology research.
> > 2. **MYC-perturbation multi-omics dataset (Proprietary)**: A specialized dataset integrating eNET-seq RNAPII occupancy, bulk RNA-seq expression, and TRRUST TF–target interactions under MYC activation/inhibition. This represents a somewhat niche, high-dimensional regulatory dataset not commonly encountered in LLM training corpora.
> >
> > **Results:** We presented a detailed analysis in `App A2`. In both domains, the system successfully adapted its tool usage and research behaviors without architectural changes, generating several domain-plausible and non-obvious findings:
> > * **NHANES — Acidosis × Insulin-Resistance Interaction**: ASCollab identified a synergistic interaction between low serum bicarbonate and high TyG index in elevating MI risk, with serum albumin moderating this effect, surfacing a cross-metabolic interaction not typically examined in standard epidemiologic workflows.
> > * **NHANES — Liver–Kidney Metabolic Flexibility Marker:** The system proposed the AST/Creatinine ratio as a protective marker of metabolic flexibility with a non-linear effect, particularly in metabolic-syndrome subgroups, highlighting a physiologically interpretable but underutilized biomarker.

---

> > > ### Author Response · Authors · 2025-11-25
> > > **Response to Reviewer KDz9 (Part 3/3)**
> > >
> > > * **MYC-Perturbation Dataset — MYC-Aligned Anabolic Program:** ASCollab identified a coherent regulatory triad (EEF2, SLC7A5, FASN) that together constitute a MYC-driven anabolic program, integrating transcriptional, pausing, and pathway data into a unified mechanistic model.
> > >
> > > **Takeaway.** When considered along with exisitng results, we have demonstrated `ASCollab` operating over seven distinct modalities: RNA-Seq, proteomics, pathway/topology data, drugs & perturbations databases, Net-Seq, and population-level epidemiology. This breadth highlights that the system is **modality-agnostic, and its social architecture transfers effectively to new scientific regimes**.
> > >
> > > **Note #1.** Due to limited expert availability during the rebuttal period, we provide qualitative domain-expert assessments rather than full rubric-based scoring. Nevertheless, in both datasets ASCollab produced novel, high-quality, and domain-plausible hypotheses. We will include expanded expert evaluations in the camera-ready version.
> > >
> > > **Note #2.** Our choice of additional evaluations is constrained by domain expertise, and we acknowledge that `ASCollab` has not yet been systematically tested in non-biological domains. While such studies are difficult to conduct within the rebuttal period, we now explicitly highlight this as an important direction for future work. We remain confident that the framework’s core principles are broadly domain-agnostic.
> > >
> > > ---
> > >
> > > ### [P4] Consideration of Different LLMs
> > > We appreciate the reviewer’s questions regarding model dependence and the role of stronger LLM backbones. Our new ablation experiments (detailed in **[P2]** and `App A1`) directly address these concerns:
> > >
> > > 1. **Strong single-agent Baseline (GPT-5).** We ran the single-agent baseline using GPT-5, providing a substantially stronger individual model.
> > >
> > > While the single agent produced some higher-quality individual reports, it still falls short of the `ASCollab` on diversity and cumulative novelty. The single agent converges on a narrower conceptual region. In contrast, the network continues to explore multiple mechanistic directions in parallel and cross-pollinates findings, resulting in higher-quality and more diverse findings.
> > >
> > > **Takeaway.** While having stronger individual agents in `ASCollab` will further enhance scientific outcomes, the social dynamics play a crucial role in organizing individual agent capabilities and effectively leveraging the heterogeneity, divergent exploration and expertise of strong individual agents.
> > >
> > > 2. **Heterogeneous LLM Backends**: We instantiated ASCollab with a mixed population of agents powered by GPT-5, GPT-4o, DeepSeek-3.1, and Llama-3.3.
> > >
> > > The heterogeneous-backbone `ASCollab` achieved the strongest performance on the novelty–quality–diversity frontier across all runs. **Model diversity amplified epistemic heterogeneity**, leading to broader exploration and richer collaboration patterns (illustrated through the analyses in `Fig 7`).
> > >
> > > **Takeaway.** This demonstrates that the framework generalizes to and can benefit from mixing LLMs (to boost heterogeneity further) rather than relying on a single powerful backbone.
> > >
> > > ---
> > >
> > > *Thank you again for the careful and insightful evaluation. We hope our responses have fully addressed your concerns, and we would be glad to clarify or expand on any remaining points. We appreciate your time and consideration.*

---

### Author Response · Authors · 2025-12-03
**Summary of Responses to Reviewer Feedback**

*We thank the reviewers for their detailed considerations and constructive comments.*

We are encouraged that reviewers found the “vision compelling,” highlighting our “ambitious and creative idea” of modeling discovery as a social process (`KDz9`), describing the framework as “comprehensive and logically coherent” (`zfHW`), and recognizing that the work “fills a critical gap” (`pLNm`). Reviewers also emphasized the potential impact of ASCollab, noting its ability to generate “diverse, novel, and scientifically plausible hypotheses” (`pLNm`) and to act as a “collaborative catalyst for bridging data-to-knowledge gaps” (`rEBh`).

We are pleased that they valued the experimental depth, pointing to “concrete downstream tasks” and rediscoveries of known biomarkers (`zfHW`, `rEBh`), as well as the system’s “emergent behaviors and self-evolving knowledge flows” (`rEBh`). Finally, we appreciate reviewers’ positive remarks on clarity and presentation, with the writing described as “easy to follow” (`pLNm`).

We have also taken the reviewers’ feedback into account. Below, we summarize our key responses to *shared concerns across reviewers*:

| **Reviewer Concern** | **Reviewers** | **Our Response** |
|---|---|---|
| **A. Clarifying conceptual novelty** | `KDz9`, `zfHW`, `pLNm` | Clarified the **core conceptual contribution**: scientific discovery as an emergent social process driven by systemic heterogeneity in agents, dynamic collaboration networks, and shared social memory. |
|  |  | Refined writing to clearly distinguish AScience from prior convergent, task-driven, or hierarchical multi-agent systems. |
| **B. Empirically validating the contribution of social dynamics** | `KDz9`, `pLNm`, `zfHW`, `rEBh` | Added a **comprehensive ablation suite** isolating the roles of collaboration, heterogeneity, and social memory. **New results** (`Fig. 6a–c`, `Fig. 7`, `Fig. 8`; `App A1`). |
|  |  | Showed that **ASCollab consistently outperforms** all ablations in expert-judged tournaments, with collaboration improving quality/novelty, heterogeneity driving diversity, and social memory enabling cumulative refinement. |
| **C. Extending evaluation to additional scientific domains** | `KDz9`, `pLNm`, `zfHW`, `rEBh` | Added **two new datasets** (NHANES epidemiology, MYC perturbation multi-omics). **New results** (`App A.2`). |
|  |  | Demonstrated cross-domain generality across **5 problems** and **7 biomedical modalities**, supported by domain-expert assessments confirming plausibility and significance. |
| **D. Integrating diverse LLM backbones** | `zfHW`, `KDz9` | Added experiments with **heterogeneous LLM backbones** (GPT-5, GPT-4o, DeepSeek-3.1, Llama-3.3). **New results** (`Fig. 6`). |
|  |  | Found that backbone diversity enhances epistemic heterogeneity, yielding richer exploratory behavior and improving novelty–quality–diversity frontier performance. |
| **E. Clarifying problem scope** | `zfHW`, `KDz9` | Refined the Introduction and Sec. 2.2 to more clearly define **hypothesis hunting** as a distinct problem setting. |
|  |  | Emphasized its defining characteristics: **open-ended exploration**, **cumulative refinement**, and **dataset-defined, endogenously evaluated hypotheses**. |


We additionally provide detailed point-by-point responses to each reviewer in the individual rebuttals. We hope these updates fully address the reviewers’ concerns.


*With thanks,*

The Authors of #20843

---

### Meta-Review · Area_Chair_65u5 · 2026-01-03

**Summary:**

The paper introduces a multi-agent framework for open-ended “hypothesis hunting,” modeling scientific discovery as a social process among heterogeneous LLM agents.

Reviewers found the vision ambitious and appreciated the system’s ability to generate diverse, plausible hypotheses within cancer genomics. However, concerns centered on limited conceptual novelty, as much of the system extends established multi-agent orchestration paradigms without introducing fundamentally new algorithmic principles. The empirical scope was also viewed as narrow, with key evaluations confined to TCGA and requiring subjective expert judgments. Even with substantial additions in rebuttal, the work remains more of an impressive engineering system than a research contribution meeting the standards of ICLR.

**Reviewer Concerns:**

The rebuttal effectively addressed many surface-level issues, including broader datasets, added ablations, heterogeneous backbones, and improved clarity on terminology and experimental setups. These additions strengthened the empirical narrative but did not resolve core concerns shared across reviewers. Specifically, the methodological contribution remains incremental, and the conceptual distinction between “hypothesis hunting” and standard hypothesis generation was not sufficiently convincing. The system’s scientific insights rely heavily on subjective expert evaluation, and the work lacks clear evidence that the social-agentic layer meaningfully advances discovery beyond what strong single-model baselines or simpler coordination structures could achieve. These unresolved concerns fundamentally limit the perceived novelty and rigor of the contribution.

**Reviewer Scores:**

Reviewer KDz9: Likely unchanged (2). Their concerns about novelty and comparative baselines remain largely unaddressed.

Reviewer pLNm: Possibly unchanged or slightly lowered (6, or from 6 to 4), as added domains help, but core issues about evaluation robustness and scalability persist.

Reviewer zfHW: Likely unchanged (4), concerns regarding novelty and clear distinction from prior work remain.

Reviewer rEBh: Likely unchanged (6). Although positive, the reviewer’s stated limitations (domain specificity, computational cost, and lack of theoretical grounding) are only partially mitigated.

---

### Decision · Program_Chairs · 2026-01-26

Reject